# The distribution of antibiotic use and its association with antibiotic resistance

Scott W Olesen[1], Michael L Barnett[2,3], Derek R MacFadden[4], John S Brownstein[5,6], Sonia Hernández-Díaz[7], Marc Lipsitch[1,7,8], Yonatan H Grad[1,9]*

[1]Department of Immunology and Infectious Diseases, Harvard T.H. Chan School of Public Health, Boston, United States; [2]Department of Health Policy and Management, Harvard T.H. Chan School of Public Health, Boston, United States; [3]Division of General Internal Medicine and Primary Care, Department of Medicine, Brigham and Women's Hospital, Harvard Medical School, Boston, United States; [4]Division of Infectious Diseases, Department of Medicine, University of Toronto, Toronto, Canada; [5]Boston Children's Hospital, Boston, United States; [6]Harvard Medical School, Boston, United States; [7]Department of Epidemiology, Harvard T.H. Chan School of Public Health, Boston, United States; [8]Center for Communicable Disease Dynamics, Harvard T.H. Chan School of Public Health, Boston, United States; [9]Division of Infectious Diseases, Department of Medicine, Brigham and Women's Hospital, Harvard Medical School, Boston, United States

*For correspondence:
ygrad@hsph.harvard.edu

**Abstract** Antibiotic use is a primary driver of antibiotic resistance. However, antibiotic use can be distributed in different ways in a population, and the association between the distribution of use and antibiotic resistance has not been explored. Here, we tested the hypothesis that repeated use of antibiotics has a stronger association with population-wide antibiotic resistance than broadly-distributed, low-intensity use. First, we characterized the distribution of outpatient antibiotic use across US states, finding that antibiotic use is uneven and that repeated use of antibiotics makes up a minority of antibiotic use. Second, we compared antibiotic use with resistance for 72 pathogen-antibiotic combinations across states. Finally, having partitioned total use into extensive and intensive margins, we found that intense use had a weaker association with resistance than extensive use. If the use-resistance relationship is causal, these results suggest that reducing total use and selection intensity will require reducing broadly distributed, low-intensity use.
DOI: https://doi.org/10.7554/eLife.39435.001

## Introduction

Antibiotic use is a primary driver of antibiotic resistance, and reducing antibiotic use is a central strategy for combatting resistance (*Gould, 1999*; *Harbarth and Samore, 2005*). Understanding the relationship between antibiotic use and antibiotic resistance is therefore critical for the design of rational antibiotic stewardship strategies. Multiple studies have identified cross-sectional relationships between antibiotic use and resistance, especially across European countries and US states (*Goossens et al., 2005*; *García-Rey et al., 2002*; *Bronzwaer et al., 2002*; *Bell et al., 2014*; *European Centre for Disease Prevention and Control et al., 2017*; *van de Sande-Bruinsma et al., 2008*; *MacFadden et al., 2018a*). In general, these studies compare total outpatient antibiotic use with population-level resistance. However, antibiotic use is generally not evenly distributed. A study of outpatient prescribing in the UK found that 30% of patients were prescribed at least one antibiotic per year, with the top 9% of patients receiving 53% of all antibiotics (*Shallcross et al., 2017*). A

study of beneficiaries of Medicare, a national health insurance program that covers that vast majority of Americans 65 and older, found that the proportion of beneficiaries who take antibiotics varies by US state and drug class (*Zhang et al., 2012*). In some cases, antibiotic courses can last for months or even years (*Enzler et al., 2011*; *Lau et al., 2017*). Because antibiotic use is uneven, total use does not distinguish between broad use—many people receiving a few prescriptions—and intense use—a few people receiving many prescriptions (*Berrington, 2010*).

It stands to reason that the distribution of antibiotic use, not just total use, could have an effect on resistance (*Turnidge and Christiansen, 2005*). There are few studies of the relationship between repeated antibiotic exposure on antibiotic resistance (*Costelloe et al., 2010*; *Hillier et al., 2007*; *Carothers et al., 2007*; *Arason et al., 1996*; *Nasrin et al., 2002*; *McMahon et al., 2003*; *Catry et al., 2018*), and it remains unclear whether broad use or intense use is associated with population-level resistance. For example, if a first course of antibiotics given to an antibiotic-naive patient clears most of the susceptible bacteria they carry, then a second course in the same patient will have only a small effect, since most susceptible bacteria were already eliminated. Giving that second course to a different, antibiotic-naive patient instead would have a greater effect on population-level resistance. On the other hand, multiple courses given to a single patient might have a synergistic effect on resistance.

The goal of this study was to test the hypothesis that intense antibiotic use has a stronger association with population-level resistance than broad, low-level antibiotic use. We used an ecological design to compare the distribution of antibiotic use with antibiotic resistance. Although an ecological design is potentially subject to confounders and cannot definitively test for the causal effect of the distribution of use on resistance at the individual level, ecological studies of use and resistance are the most feasible design for studying the relationship between antibiotic use and population-level resistance, and the results of ecological designs play an important role in developing antibiotic stewardship policies (*Huttner and Samore, 2011*; *Schechner et al., 2013*).

To test this hypothesis, we first characterized the distribution of outpatient antibiotic use in two US nationwide pharmacy prescription claims databases, Truven Health MarketScan Research Database (*Truven Health Analytics, 2015*) and Medicare, both covering 2011–2014. We considered only outpatient antibiotic prescribing, which accounts for 80–90% of total medical antibiotic use in the UK and Sweden (*Public Health Agency of Sweden and National Veterinary Institute, 2015*; *ESPAUR Writing Committee, 2014*) and is presumed to account for a similar fraction in the US (*Centers for Disease Control and Prevention, 2017*). Unlike antibiotic sales data and nationwide healthcare surveys (*Hicks et al., 2015*), MarketScan and Medicare claims data, which have previously been used to characterize variations in antibiotic use (*Zhang et al., 2012*; *Suskind et al., 2016*; *Owusu-Edusei et al., 2015*; *Arizpe et al., 2016*), provide longitudinal prescribing information about individual people, and can therefore distinguish between many people getting a few prescriptions and a few people getting many prescriptions. We characterized the distribution of antibiotic use across US states by partitioning annual total use as the sum of annual first use—individuals' first pharmacy fill for an antibiotic in a calendar year—and annual repeat use—pharmacy fills beyond individuals' first ones in a calendar year. Second, we compared annual total antibiotic use with antibiotic resistance as measured in ResistanceOpen, a US nationwide sample of antibiotic susceptibility reports, for 2012–2015 (i.e. lagged by 1 year (*van de Sande-Bruinsma et al., 2008*; *Bruyndonckx et al., 2015*)), evaluating the relationship between use and resistance across US states for 72 pathogen-antibiotic combinations. Finally, we evaluated whether annual first use and annual repeat use are differently associated with population-level resistance.

## Results

### Antibiotic use is not evenly distributed

Our analysis included 99.8 million outpatient pharmacy antibiotic prescription fills among 62.4 million unique people, approximately 20% of the US population, during 2011–2014 using the MarketScan database (*Truven Health Analytics, 2015*). In 2011, 34% of people received an antibiotic, and 10% of people received 57% of all antibiotic prescriptions. This distribution varied by population but was similar across data years (*Figure 1—figure supplement 1*). To characterize the distribution of specific antibiotics, we grouped individual antibiotic generic formulations into drug groups based on

their chemical structures and mechanisms of action (*Supplementary file 1* - Table 1). For all drug groups, most people had zero prescriptions for that antibiotic in a given year, but antibiotics differed in their distributions (*Figure 1*).

We next examined the distribution of antibiotic use for each drug group and US state. To quantify the distribution of antibiotic use, we labeled each antibiotic pharmacy claim as 'first' if it was the first pharmacy fill for that drug group made by that individual in that calendar year, and 'repeat' if it was a second, third, etc. fill for an antibiotic in the same drug group made by the same individual in the same calendar year. An individual's first and repeat claims in a calendar year add up to their total number of claims for that year. We then partitioned population-level annual total use, measured as pharmacy fills per 1000 members per year, into the sum of annual first use, measured as first fills per 1000 members per year, and repeat use, measured as repeat fill per 1000 members per year, for each drug group and US state. Annual first use of a drug group is equivalent to the proportion of the population taking an antibiotic in that group in that year.

Total use varied between drug groups and across states (*Figure 2*). Annual repeat use made up a steady one-quarter to one-third of annual total use across drugs and states, with the exception of tetracyclines, for which high repeat use was associated with young adults (*Figure 2—figure supplements 1* and *2*), probably for acne treatment. This distribution of first and repeat use is distinct from the pattern predicted by the single-parameter Poisson and geometric distributions (*Figure 2*), but the ratio of first use to repeat use for each drug was nearly constant across US Census regions (*Figure 2—figure supplement 3*). Thus, the higher antibiotic use in the Southern states (*Zhang et al., 2012*; *Hicks et al., 2013*) is primarily attributable to a greater proportion of people taking antibiotics, not because those who receive antibiotics receive more of them.

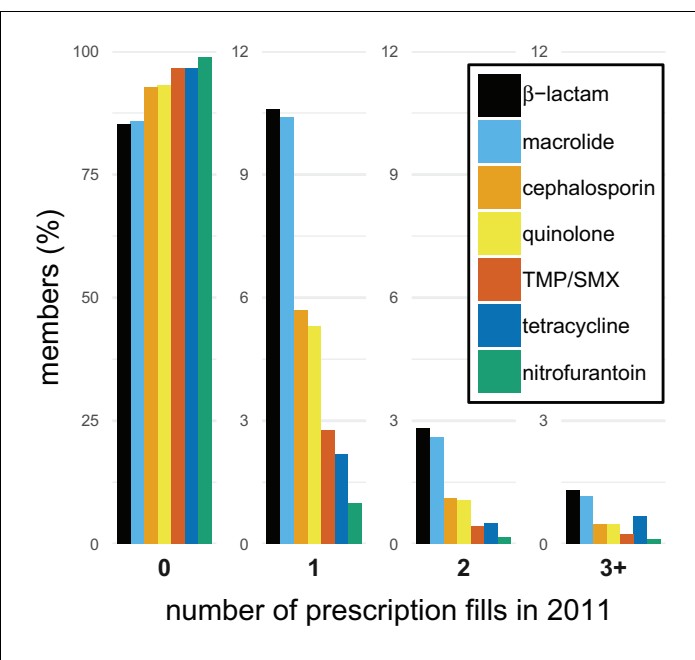

**Figure 1.** The distribution of antibiotic use within individuals. Bars indicate the proportion of members in the MarketScan data with different numbers of prescription fills in 2011 for each of the drug groups. TMP/SMX: trimethoprim/sulfamethoxazole.

DOI: https://doi.org/10.7554/eLife.39435.002

The following figure supplement is available for figure 1:

**Figure supplement 1.** Cumulative distribution of antibiotic use.

DOI: https://doi.org/10.7554/eLife.39435.003

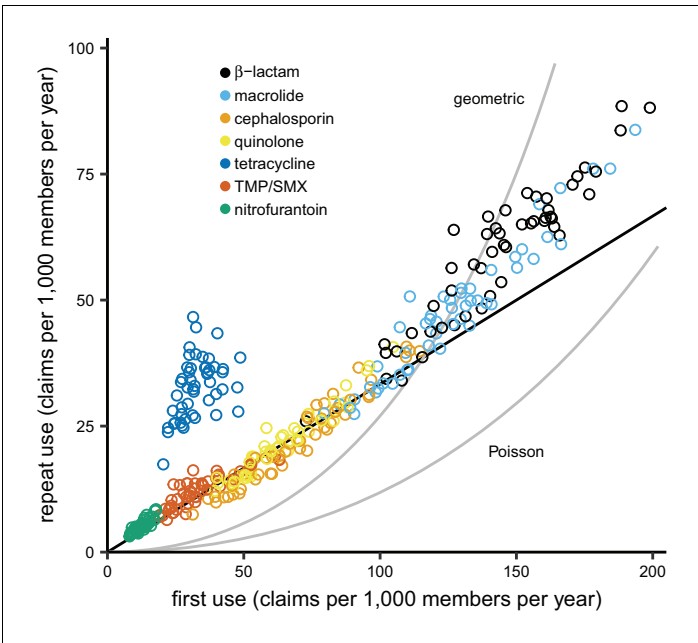

**Figure 2.** The distribution of antibiotic use across US states. Each point indicates first use and repeat use of a single drug group in a single US state (averaged over the data years). Points falling on the black line have three times as much first use as repeat use (i.e. repeat use is one-quarter of total use). The curves show the relationships between first use and repeat use expected from the Poisson and geometric distributions. TMP/SMX: trimethoprim/sulfamethoxazole.

DOI: https://doi.org/10.7554/eLife.39435.004

The following figure supplements are available for figure 2:

**Figure supplement 1.** Distribution of tetracycline use by age.
DOI: https://doi.org/10.7554/eLife.39435.005
**Figure supplement 2.** Distribution of antibiotic use by population.
DOI: https://doi.org/10.7554/eLife.39435.006
**Figure supplement 3.** Distribution of antibiotic use by region.
DOI: https://doi.org/10.7554/eLife.39435.007

## Landscape of correlations between total use and resistance across pathogens and antibiotics

To verify that our antibiotic use and resistance data sources could be used to distinguish the associations of first use and repeat use with antibiotic resistance, we first measured the landscape of Spearman correlations between total use and antibiotic resistance for multiple pathogens and antibiotics (*Goossens et al., 2005*; *García-Rey et al., 2002*; *Bronzwaer et al., 2002*; *Bell et al., 2014*; *European Centre for Disease Prevention and Control et al., 2017*; *van de Sande-Bruinsma et al., 2008*). To measure antibiotic resistance, we used ResistanceOpen, a US nationwide sample of hospital antibiotic susceptibility reports (*MacFadden et al., 2016*), which included resistance of 38 pathogens to 37 antibiotics in 641 antibiotic susceptibility reports from 230 organizations (hospitals, laboratories, and surveillance units) spread over 44 US states. Although most organizations contributing antibiotic susceptibility reports were hospitals, hospital antibiotic susceptibility reports are biased toward community-acquired organisms (*Wang et al., 2017*; *Hindler and Stelling, 2007*), community antibiotic use can drive antibiotic resistance measured in hospitals (*Knight et al., 2018*; *MacFadden et al., 2018b*), and studies often compare hospital antibiotic susceptibility reports with community antibiotic use (*Goossens et al., 2005*; *MacDougall et al., 2005*; *Hicks et al., 2011*).

Because the epidemiology and pharmacology of each pathogen-antibiotic combination is unique, each combination could have a unique use-resistance relationship (*Turnidge and Christiansen, 2005*). We therefore aggregated antibiotic resistance into the same drug groups with which we aggregated antibiotic use (*Supplementary file 1* – Table 1) and evaluated the 72 pathogen-

antibiotic combinations that were adequately represented in the antibiotic resistance data (see Materials and methods). Across those 72 combinations, correlation coefficients ranged from –32% to 64% (*Figure 3*, *Supplementary file 1* - Table 2). The strongest correlation (Spearman's $\rho$ = 64%, 95% CI 41% to 80%) was between macrolide use and the proportion of *Streptococcus pneumoniae* isolates that were macrolide nonsusceptible (*Figure 4*). Correlation coefficients were mostly positive (median correlation coefficient 21%, IQR 8% to 34%). Use-resistance correlations involving macrolides, quinolones, and cephalosporins were more positive than those for nitrofurantoin, and correlations involving quinolones were more positive than those for trimethoprim/sulfamethoxazole (pairwise Mann-Whitney tests, two-tailed, FDR = 0.05). Coefficients were not significantly more positive for any particular pathogen.

Because isolates from older adults are disproportionately represented in antibiotic susceptibility reports (*Kanjilal et al., 2018*), we suspected that population-wide resistance might, in some cases, correlate better with antibiotic use among older adults. We therefore queried outpatient pharmacy antibiotic claims records from individuals 65 and older on Medicare (see Materials and methods). When antibiotic use among Medicare beneficiaries was substituted for antibiotic use as measured in the MarketScan data (*Figure 2—figure supplement 2*), correlation coefficients were similar (*Supplementary file 1* – Tables 2 and 3). Conversely, children are the primary carriers for some pathogens (e.g. *Streptococcus pneumoniae* (*García-Rodríguez and Fresnadillo Martínez, 2002*)), so we suspected that resistance might, in other cases, better correlate with children's antibiotic use. Restricting the antibiotic use data to members at most 15 years old (*Figure 2—figure supplement 2*) again yielded similar coefficients (*Supplementary file 1* - Tables 2 and 3). Thus, the landscape of correlations we observed was mostly robust to the exact population in which antibiotic use was measured.

We also evaluated the sensitivity of the results to the measurement of antibiotic use, substituting days supply of antibiotic for number of pharmacy fills, and the geographic level of the analysis, by aggregating the Medicare use data and resistance data at the level of the 306 hospital referral regions that approximate regional health care markets (*The Center for the Evaluative Clinical Sciences, Dartmouth Medical School, 1996*) (*Supplementary file 1* – Tables 2 and 3). The absolute values of the correlation coefficients were slightly closer to zero when using days supply rather than fills (Wilcoxon test, two-tailed; pseudomedian difference in absolute correlation coefficient 1.9 percentage points, 95% CI 0.72 to 3.1) and substantially closer to zero when using hospital referral regions rather than states as the units of analysis (6.1 percentage points, 95% CI 3.0 to 9.1).

We finally evaluated the sensitivity of the results to the exact antibiotic resistance data (*Figure 3—figure supplement 1*, *Supplementary file 1* – Tables 2 and 3). First, we excluded all antibiotic susceptibility reports whose annotations indicated they include isolates exclusively from inpatient settings. Most reports include isolates from a mix of settings, so this analysis excluded only 7% of the total number of isolates from the data. Second, we included only reports whose annotations indicated their isolates were exclusively from outpatient or emergency room settings, which retained only 14% of the isolates. Correlations using the resistance data excluding inpatient-only isolates were slightly stronger than the baseline data (Wilcoxon test, two-tailed; pseudomedian difference in absolute correlation coefficient 1.1 percentage points, 95% CI 0.01 to 2.2), but correlations computed using the outpatient-only data were not systematically stronger or weaker.

## Lack of evidence for more positive association with repeat use

Having examined the landscape of the relationships between total use and resistance across pathogen-antibiotic combinations, we set out to test the hypothesis that repeat use has a stronger association with resistance than first use. For each pathogen-antibiotic combination, we performed a multiple regression predicting proportion nonsusceptible from first use and repeat use (*Figure 5*). First use and repeat use are highly correlated in some cases (*Supplementary file 1* - Table 4) which will widen the confidence intervals on the regression coefficients but should not introduce bias (*Schisterman et al., 2017*). Regression coefficients for first use were more often positive than negative (54 of 72 [75%]; binomial test, 95% CI 63% to 84%). That is, first use was positively associated with resistance when controlling for repeat use. In contrast, regression coefficients for repeat use were more often negative than positive (44 of 72 [61%]; binomial test, 95% CI 49% to 72%). That is, repeat use was negatively associated with resistance when controlling for first use.

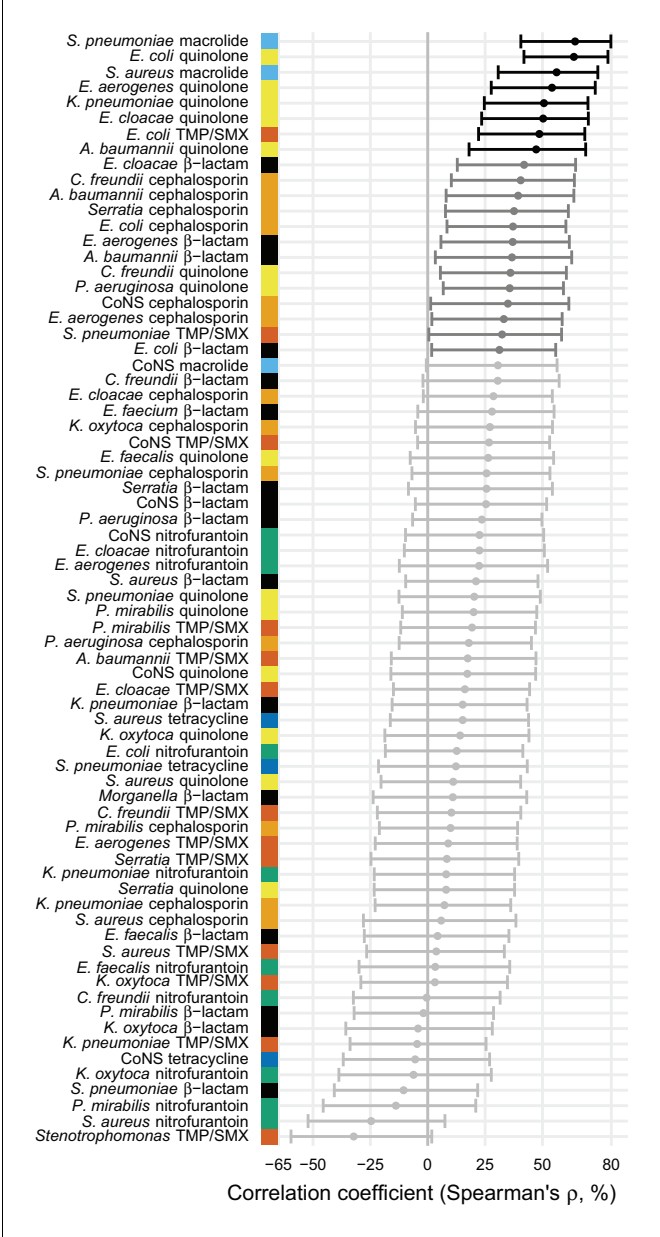

**Figure 3.** Correlations between total antibiotic use and resistance are biased toward positive values. Error bars show 95% confidence intervals. The color strip visually displays the drug groups. Statistical significance is indicated by color of the points (black, significant at FDR = 0.05, two-tailed; dark gray, significant at α = 0.05, two-tailed; light gray, not significant). TMP/SMX: trimethoprim/sulfamethoxazole. CoNS: coagulase-negative *Staphylococcus*.
DOI: https://doi.org/10.7554/eLife.39435.008

The following source data and figure supplement are available for figure 3:

**Source data 1.** Antibiotic use data.
DOI: https://doi.org/10.7554/eLife.39435.010
**Source data 2.** Antibiotic resistance data.
DOI: https://doi.org/10.7554/eLife.39435.011
**Figure supplement 1.** Correlations between total antibiotic use and resistance using subsets of the resistance data.
DOI: https://doi.org/10.7554/eLife.39435.009

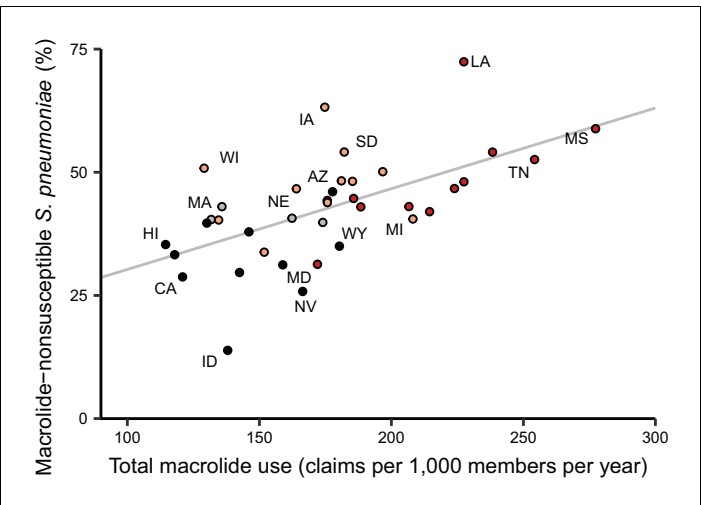

**Figure 4.** Total macrolide use and macrolide resistance among *Streptococcus pneumoniae* correlate across US states. Labels indicate selected states. Colors indicate US Census region (red, South; light red, Midwest; gray, Northeast; black, West). Line shows unweighted linear best fit. Southern states have highest macrolide use and resistance.

DOI: https://doi.org/10.7554/eLife.39435.012

We evaluated the sensitivity of this result to age group, data source, metric of antibiotic use, geographic unit of analysis, and subset of the resistance data as described above. In all cases, regression coefficients for first use in the multiple regression were more likely to be positive than negative, and regression coefficients for repeat use were more likely to be negative than positive in all but one case (*Supplementary file 1* - Table 5). For certain pathogens and antibiotics, resistance could presumably accumulate in an individual over many years (*Carothers et al., 2007*; *McMahon et al., 2003*), so we also computed alternate measures of first and repeat use by considering only individuals who were included in the MarketScan data for each year of 2011–2014, and we labeled an

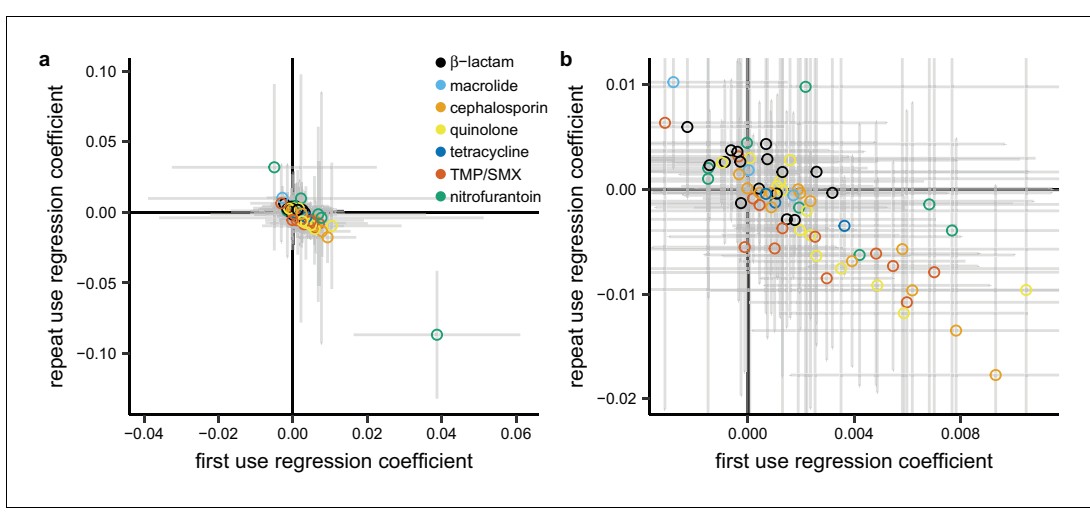

**Figure 5.** Repeat use tends to be negatively associated with resistance when controlling for first use. Each point represents a pathogen-antibiotic combination. The position of the point shows the two coefficients from the multiple regression. The units of the coefficients are proportion resistant per annual claim per 1000 people. Color indicates drug group. Error bars show 95% CIs. (**a**) All data. (**b**) Same data, showing only the center cluster of points.

DOI: https://doi.org/10.7554/eLife.39435.013

antibiotic fill as first use only if it was the first fill for that drug group made by that individual in the entire 4-year period. In that analysis, a similar proportion of regression coefficients for first use were positive (69%, 95% CI 57% to 80%) and regression coefficients for repeat use were equally likely to be positive or negative (53%, 95% CI 41% to 65%).

## Discussion

### Landscape of use-resistance relationships

We used US nationwide datasets measuring antibiotic use in 60 million individuals and antibiotic resistance in 3 million bacterial isolates to analyze relationships between antibiotic use and resistance, examining 72 pathogen-antibiotic combinations simultaneously, using identical data sources and analytical methods across combinations. Although previous studies have examined multiple pathogen-antibiotic combinations, usually no more than five pathogens or antibiotics are considered at once (*Goossens et al., 2005*; *García-Rey et al., 2002*; *van de Sande-Bruinsma et al., 2008*). We found that correlations between total use and resistance were mostly positive and that certain drugs tended to have more positive correlations, but that there was no clear pattern by organism (*Bell et al., 2014*). The overall landscape of correlations was mostly robust to the age groups studied, the subset of resistance data used, and the geographic scale of the analysis, although correlations were somewhat weaker when conducting analysis at smaller geographic scales (*van de Sande-Bruinsma et al., 2008*; *Priest et al., 2001*; *Magee et al., 1999*; *Livermore et al., 2000*). We used outpatient antibiotic use as the predictor of resistance because 80–90% of antibiotic use occurs in the outpatient setting (*Centers for Disease Control and Prevention, 2017*) and because most antibiotic pressure on pathogens is due to 'bystander selection', in which the patient is treated for some reason other than an infection caused by that pathogen (*Tedijanto et al., 2018*).

The correlations we observed between total antibiotic use and population-wide antibiotic resistance were noticeably weaker than those in highly cited European studies but comparable to those from other analyses of European data. For example, for *S. pneumoniae* and macrolides, *Goossens et al. (2005)* reported a Spearman's $\rho$ of 83% and *García-Rey et al. (2002)* reported 85%, while we found 62% and *van de Sande-Bruinsma et al. (2008)* reported a median of 56%. These studies all used similar statistical methods, so the differences in the results must be due to some other factors (e.g. data quality, range of antibiotic use, distribution of antibiotic use, or pathogen biology). Correlations between *E. coli* resistance and use of β-lactams, cephalosporins, trimethoprim/sulfamethoxazole, and quinolones were similar to those reported in studies of use and resistance in UK primary care groups (*Priest et al., 2001*; *Magee et al., 1999*).

The most notable difference, however, between our study and previous results from Europe is for *S. pneumoniae* and β-lactams: *Goossens et al. (2005)* report a Spearman's $\rho$ of 84%, but we found no relation (–11%, 95% CI –41% to 22%). We propose that the narrow variation in β-lactam use across US states, approximately two-fold between the highest- and lowest-using states, obscures a correlation that is more apparent in Europe, where there is a four-fold variation between the highest- and lowest-using countries (*Goossens et al., 2005*). Thus, our results and those from Goossens *et al.* may be consistent with respect to the underlying biology. We also note that, when reproducing the methodology from a US study (*Hicks et al., 2011*) of the use-resistance relationship for β-lactams and *S. pneumoniae* (dichotomizing states as high- or low-prescribing and computing the odds ratio of resistance), we find a consistent point estimate but with wider confidence intervals (1.15, 95% CI 0.75 to 1.76).

Our study design may limit the interpretability of the landscape of use-resistance relationships. First, like the leading European studies using EARS-Net and US studies using the Centers for Disease Control and Prevention Active Bacterial Core surveillance, we compare population-wide outpatient antibiotic use with antibiotic susceptibility reports from hospitals. The degree to which hospital antibiotic susceptibility reports represent community infections is debated (*Wang et al., 2017*; *MacDougall et al., 2005*). For example, if outpatient antibiotic use selects for resistance among community-acquired infections, and hospital antibiograms reflect data from community-acquired infections as well as unrelated inpatient resistance patterns, then the correlations we measure would be biased toward weaker associations. Furthermore, antibiotic use and resistance in the community

setting is not completely independent of use and resistance in the hospital setting (**Knight et al., 2018**), and our approach does not account for any relationship between the two.

Second, antibiotic resistance is temporally dynamic, and our cross-sectional approach assumes that antibiotic use is autocorrelated across years (**Lipsitch, 2001**) or that resistance changes slowly (**Hennessy et al., 2002**). If use does cause resistance, and use and resistance changed meaningfully over the course of the study, then the correlations we measured by aggregating over all years would be biased toward weaker associations.

Third, because of the limitations in statistical power, we did not address the possibility that use of one antibiotic can select for resistance to another antibiotic (**Pouwels et al., 2018**; **Weber et al., 2003**). Notably, use of one antibiotic can select for resistance to another antibiotic if the dominant clones of that species are resistant to both (**Pouwels et al., 2018**). In that case, if the use rates of the two drugs are correlated across states, then the apparent relationship between one drug and resistance to that drug would be biased upward. Furthermore, because the palette of antibiotic use varies by country (**Van Boeckel et al., 2014**), and different pathogen strains circulate in different populations, the univariate associations we observed between use of an antibiotic and resistance to that antibiotic in the US may not be applicable in other geographies.

Finally, like in other studies of antibiotic use, we did not address patient adherence, and typical approaches to address adherence using claims data (**Steiner and Prochazka, 1997**) are problematic when the intended duration of treatment is not clear. The measured correlation would then be biased if, for example, poor patient adherence increased resistance and patient adherence were also correlated with antibiotic use.

## Distribution of antibiotic use and antibiotic resistance

We described the distribution of antibiotic use across drug groups and US states, finding that 34% of the study population took an antibiotic in a year, and 10% of the population had 57% of the antibiotic fills in that year, similar to results from the UK (**Shallcross et al., 2017**), although this distribution varied by population (**Figure 1—figure supplement 1**). By partitioning annual total use into annual first use and annual repeat use, we were able to show that, for each drug, annual first use makes up the majority of annual total use and that variations in annual first use explain more variance in annual total use than do variations in annual repeat use. We also found that first use tends to have a positive association with resistance when controlling for repeat use, while repeat use tends to have negative associations with resistance when controlling for first use. This result held across sensitivity analyses.

If these associations are causal, that is, if outpatient first and repeat antibiotic use select for resistance among community-acquired pathogens, then our results would imply that antibiotic resistance in the outpatient setting is due more to first use, which tended to have positive associations with resistance, than to repeat use. In contrast to proposals to focus on intense antibiotic users for combating resistance (**Shallcross et al., 2017**), this situation would imply that preventing marginal prescriptions among patients whose indications are borderline-appropriate or inappropriate for antibiotics may be the more effective tactic for reducing the prevalence of resistance mechanisms already established in the US.

There are limitations to the interpretability of these results. First, as mentioned above, there is a potential mismatch between the sources of the antibiotic use and antibiotic resistance data.

Second, although antibiotic use is a major driver of antibiotic resistance, the observed results may not be causal. Factors beyond antibiotic use, like population density, play a role in antibiotic resistance (**Harbarth and Samore, 2005**; **MacFadden et al., 2018a**). Even if antibiotic use and resistance are causally related, it may be that resistance affects antibiotic use. For example, if resistance to a drug is high, treatment using that drug is more likely to fail, discouraging repeated use, so that high resistance lead to decreased repeat use (**Wang et al., 2017**; **Priest et al., 2001**; **Pouwels et al., 2018**). Ecological studies like this one do not directly address causality, and further work is needed to distinguish between different causal pathways.

Third, the observed population-level relationships between antibiotic use and resistance need not also hold for the relationship between an individual's first and repeat antibiotic use and the risk of a resistant infection in that individual. Any comparisons between our population-level results and individual-level studies would need to account for the difference between our population-level measures

of first and repeat antibiotic use and the individual-level timing of antibiotic use and measurements of resistance.

Fourth, controlling for factors beyond antibiotic use could alter the apparent relationship between antibiotic use and resistance. In particular, we speculate that controlling for patient morbidity, which we did not address in this population-level analysis, would amplify the observed result, that first use tends to have a more positive association with antibiotic resistance than repeat use. We expect that morbid individuals have more repeat antibiotic use. We also expect that morbid individuals visit the hospital more often, putting them at higher risk of antibiotic resistant infections regardless of their antibiotic use. Thus, we speculate that repeat use causes resistance and also is a predictor of morbidity, which is associated with resistance. Failing to control for morbidity thus biases the association between repeat use and resistance toward more positive values. Conversely, controlling for morbidity would decrease the measured relationships between repeat use and resistance, amplifying our central result.

Fifth, we defined first and repeat use with respect to the calendar year, while it may be that some other timescale is the appropriate one for this analysis. Although our central result held when redefining first and repeat use with respect to a 4-year period (*Supplementary file 1* - Table 5), it may be that, say, repeat use within an individual on a time-scale shorter than a year is an important determinant for risk of resistance in that individual. Our study does not distinguish between repeat use that occurs across calendar year boundaries, which is presumably important for relating individuals' antibiotic use with their risk of resistance.

Finally, we note that first use and repeat use are only one set of many ways of measuring the distribution of antibiotic use. For example, 10 repeat uses could mean 1 person with 10 repeat uses or 10 people with 1 repeat use each. The first and repeat use metrics cannot distinguish between these two cases, and it may be that some other measure of the distribution of antibiotic use would yield different results.

In conclusion, we find that population-wide antibiotic use and population-wide resistance appears to be more closely linked with broadly distributed, low-intensity use rather than with intensity of use. Ultimately, accurate models predicting the emergence and spread of antibiotic resistance will require more careful characterizations of who gets what antibiotic (*Harbarth et al., 2001*), what selection pressure that places on pathogens, how those pathogens are transmitted, and in whom they manifest as infections (*Lipsitch and Samore, 2002*). An ideal study would compare the complete history of an individual's outpatient and inpatient antibiotic exposure with clinical microbiology data from that same individual, cross-referenced against population-level factors, among a representative, nationwide sample of individuals. Individual-level results could then also be compared with mechanistic models of resistance to draw inferences about within-host effects of antibiotic use (*Levin et al., 1997*; *Colijn et al., 2010*), and the role of co-occurring resistance and correlated antibiotic use could be addressed. In the absence of such a dataset, these ecological, associative results provide a guide to the development of antibiotic stewardship policy.

## Materials and methods

### Study population and antibiotic use

MarketScan (*Truven Health Analytics, 2015*) data covering 2011–2014 were used to identify insurance plan members and characterize their outpatient antibiotic use. To ensure quality of the antibiotic use distribution data, only members who were on their insurance plan for 12 months during a given calendar year were included. Prescription fills for oral and injected antibiotics were identified by generic formulation (*Supplementary file 1* - Table 6) and drug forms (*Supplementary file 1* - Table 7). We treated multiple claims on the same day for the same generic formulation with the same refill code as a single prescription fill. In the main analysis, antibiotic use was measured using fills, rather than days supply of drug, because some previous research has suggested that prescriptions better correlate with resistance (*Bruyndonckx et al., 2015*) and that this choice is probably not detrimental (*van de Sande-Bruinsma et al., 2008*; *Lipsitch, 2001*). Generic drugs were grouped into antibiotic drug groups designed to correspond to the antibiotic resistance drug groups described below (*Supplementary file 1* - Table 1). All measures of antibiotic use were computed for

each year 2011 to 2014, and the mean for each value across the 4 years was reported and used in analyses of the use-resistance relationship.

Antibiotic use among Medicare beneficiaries was measured as previously described (*Olesen et al., 2018*). Briefly, we considered fee-for-service beneficiaries at least 65 years old among and with 12 months of enrollment in Medicare Part B and Part D among a 20% sample of beneficiaries for each year 2011 to 2014. The Medicare data, which provides the ZIP Code for each beneficiary, were also aggregated at the level of hospital referral region (*The Center for the Evaluative Clinical Sciences, Dartmouth Medical School, 1996*), using the 2011 ZIP Code to region crosswalk. MarketScan data do not include ZIP Code-level resolution.

This study was deemed exempt from review by the institutional review board at the Harvard T. H. Chan School of Public Health.

## Antibiotic resistance

Antibiotic resistance prevalences for common bacterial pathogens were identified from ResistanceOpen, a previously developed database of spatially localized patterns of antibiotic resistance (*MacFadden et al., 2016*). This continuously updated database contains antibiotic resistance data from online sources during 2012 to 2015. At the time of analysis, the resistance data consisted of approximately 86,000 records, each indicating the fraction of isolates of an organism that were nonsusceptible to a particular drug in a particular antibiotic susceptibility report ('antibiogram'). The median number of isolates corresponding to each record was 93, but records had up to 75,000 associated isolates. Seven records (<0.01%) with missing numbers of isolates were excluded. In antibiograms that separated *S. aureus* into MRSA and MSSA, resistance of aggregate *S. aureus* to individual drugs was taken as the average of the MRSA and MSSA records, weighted by number of isolates. MRSA and MSSA were not considered as separate species in any analysis.

Antibiotics used in antibiotic resistance assays were grouped into antibiotic drug groups (*Supplementary file 1* - Table 1) designed to correspond to the antibiotic use groups. If resistance to more than one antibiotic in a drug group was reported for a particular pathogen in a particular antibiogram, resistance to that drug group for that pathogen in that antibiogram was computed as the mean of the resistances measured for the antibiotics in that group, weighted by the number of isolates. The proportion of nonsusceptible isolates in a state for a particular pathogen-antibiotic combination was computed as the average of the proportions from each contributing antibiogram in that state, weighted by number of isolates.

## Statistical methods

Antibiotic use and resistance were compared using Spearman correlations and multiple linear regressions. Of the 887 pathogen-antibiotic combinations present in the data, we analyzed the 72 combinations that were present in at least 34 states. This excluded 21% of the pathogen-antibiotic-antibiogram combinations. We established the cut-off for number of states because 80% power to detect a Pearson correlation coefficient of magnitude 0.55 at $\alpha$ = 0.01 under a two-sided hypothesis requires at least 34 samples. There is no straightforward power calculation methodology for Spearman correlations, so we used the Pearson power calculation as an approximation. We aggregated data across all years, rather than comparing use and resistance in each year, because of the sparsity of the resistance data: of 2767 pathogen-antibiotic-state combinations in the data, only 182 have data for all 4 years. No pathogen-antibiotic combination had more than 4 states with data for all 4 years. Confidence intervals on correlation coefficients were computed using the Fisher transformation and normal approximation method. Multiple comparisons were accounted for using the Benjamini-Hochberg false discovery rate (FDR) (*Benjamini and Hochberg, 1995*). Multiple regressions predicted proportion of isolates nonsusceptible from first use and repeat use.

## Acknowledgements

SWO and ML were supported by cooperative agreement U54GM088558 from the National Institute of General Medical Sciences. The content is solely the responsibility of the authors and does not necessarily represent the official views of the National Institute of General Medical Sciences or the National Institutes of Health.

## Additional information

### Competing interests

Marc Lipsitch: Reviewing editor, *eLife*. The other authors declare that no competing interests exist.

### Funding

| Funder | Grant reference number | Author |
| --- | --- | --- |
| National Institute of General Medical Sciences | U54GM088558 | Scott W Olesen Marc Lipsitch |

The funders had no role in study design, data collection and interpretation, or the decision to submit the work for publication.

### Author contributions

Scott W Olesen, Conceptualization, Formal analysis, Investigation, Visualization, Methodology, Writing—original draft, Writing—review and editing; Michael L Barnett, Conceptualization, Formal analysis, Investigation, Methodology, Writing—review and editing; Derek R MacFadden, Conceptualization, Resources, Writing—review and editing; John S Brownstein, Resources, Writing—review and editing; Sonia Hernández-Díaz, Resources, Methodology, Writing—review and editing; Marc Lipsitch, Conceptualization, Formal analysis, Supervision, Funding acquisition, Writing—original draft, Writing—review and editing; Yonatan H Grad, Conceptualization, Resources, Formal analysis, Supervision, Funding acquisition, Writing—original draft, Writing—review and editing

### Author ORCIDs

Scott W Olesen (iD) http://orcid.org/0000-0001-5400-4945

Marc Lipsitch (iD) http://orcid.org/0000-0003-1504-9213

Yonatan H Grad (iD) http://orcid.org/0000-0001-5646-1314

### Decision letter and Author response

Decision letter https://doi.org/10.7554/eLife.39435.017
Author response https://doi.org/10.7554/eLife.39435.018

## Additional files

### Supplementary files

• Supplementary file 1. Supplemental tables.
DOI: https://doi.org/10.7554/eLife.39435.014

• Transparent reporting form
DOI: https://doi.org/10.7554/eLife.39435.015

### Data availability

State-level, aggregate antibiotic use and resistance data used in the main analyses are in Figure 3 - Source data 1 and 2. We do not own and cannot publish disaggregated MarketScan or Medicare data. MarketScan data are available by commercial license from Truven Health (marketscan.truven-health.com). Medicare data are available from ResDAC (www.resdac.org). ResDAC requires an application ensuring that requesting researchers comply with Common Rule, HIPAA, and CMS security and privacy requirements. Disaggregated ResistanceOpen data are restricted due to hospitals' privacy concerns. ResistanceOpen data are available by request from HealthMap (www.resistanceopen. org).

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
