## [Decision Letter]

Thank you for submitting your article "The distribution of antibiotic use and its association with antibiotic resistance" for consideration by *eLife*. Your article has been reviewed by Prabhat Jha as the Senior Editor, a Reviewing Editor, and three reviewers. The following individual involved in review of your submission has agreed to reveal her identity: Gwenan M Knight (Reviewer #3).

The reviewers have discussed the reviews with one another and the Reviewing Editor has drafted this decision to help you prepare a revised submission.

Summary:

This paper uses observational data from the US to explore the relationship between antibiotic use and levels of resistance, showing that total volumes used predict resistance levels better than rates of repeated use in a minority of individuals.

Essential revisions:

This is a useful and important analysis of antibiotic usage and resistance patterns across US states. The authors pose a specific question as to whether first use is more closely correlated with resistance than repeat use. A strength is that they are able to explore this for a large number of drug-bug combinations.

One obvious difficulty is the definition of first and repeat use. First use per calendar year immediately looks problematic. In the Discussion section, mention is made of first use in a four-year period giving similar results, but there remains the possibility that there was previous usage prior to the observation period. It is likely that some repeat use is misclassified as first use but not vice versa. The scale of the problem could be assessed for, say, years 2, 3 and 4 by checking for usage in the previous calendar year, but this is not done.

Another obvious problem is that the usage data are for outpatients, but the resistance data are (mainly) from hospitals, and that the latter data was only a small sample of the levels of resistance. It's likely that the antibiotics used in hospitals are very different to those used in the outpatient setting and hence the resistances selected may vary. This concern is addressed well in the Discussion section, however, and we don't think there is further work that could be done apart from looking at the subset of community associated infections if those data were available. But this is a major caveat.

The key result (subsection “Lack of evidence for more positive association with repeat use”) is that usage-resistance correlations across states tended to be positive for first use and negative for repeat use. The latter result appears to be unexpected and so post hoc explanations are invoked. Care is needed here. It is clearly reasonable to suggest that first use might generate resistance. Presumably subsequent use can then only generate resistance in still susceptible cases (see Introduction). And so on. But it is not clear how that process plays out in terms of generating the correlations explored here. It would have been good to have seen the authors' expectations set out and justified (possibly using a mathematical model). That would help with interpreting the results. Certainly, the Discussion section might benefit from including something about what these conclusions mean for the modelling of resistance. For instance, from a within-host perspective, do the conclusions imply that one dose is enough to generate a resistant dominant population in a host? And that the clearance rate is low?

More generally, the statistical analyses presented are very simple. This aids transparency, but a lot more could have been looked at with more sophisticated regression modelling (never mind dynamical models). For instance, multivariable, multivariate regression applied to all bug-drug combinations simultaneously could look at drug, extensive/intensive use, bug and geographic (i.e. states) which might be resolvable by combining the data. Such an approach would have also allowed formal model selection/comparison to resolve statistically significant effects. The authors should at least comment on the potential of more advanced analyses and why they chose not to pursue these here.

In doing more to explain/unravel the trends seen, the route to resistance (e.g. mutation vs. mobile genetic element) might be considered, e.g. exploring if those with a mutational based resistance (e.g. fluoroquinolones) had stronger correlations with use – repeat use in particular.

Obviously, this is an analysis of trends in the US. In a few places it would be good to compare the US situation to other countries (e.g. in the percentage of antibiotics used in outpatient vs. healthcare settings, perhaps the distribution of antibiotics used) and to explain more in detail the population (e.g. what percentage of the population is included and what is the Medicare population?).

[Editors' note: further revisions were requested prior to acceptance, as described below.]

Thank you for resubmitting your work entitled "The distribution of antibiotic use and its association with antibiotic resistance" for further consideration at *eLife*. Your revised article has been favorably evaluated by Prabhat Jha (Senior Editor), a Reviewing Editor, and three reviewers.

The manuscript has been improved but we would still like a more effort put into addressing reviewer #2's concerns below regarding hospital vs. outpatient antibiotic use. For instance, presentation of datasets showing some geographic correlation between usage in outpatient and hospital settings.

*Reviewer #1:*

I am generally happy. The authors could have been slightly more open to looking at some of the additional analyses suggested by the reviewers in my opinion, but they make reasonable arguments about why they choose not to. The changes to the text address the issues of clarity and caveats highlighted by the reviewers.

*Reviewer #2:*

I do not think this manuscript has been improved. The authors have responded to the reviewers' comments, but have made only minor changes to the manuscript, even in response to the "essential" revisions. Those changes are largely confined to explanations, caveats and discussion; as far as I can see, they have not revised their analyses in any way.

In particular they have not been able to address in any meaningful way the mis-match between scoring antibiotic usage in outpatients but resistance in hospital patients. This seems to me to be a fundamental flaw in the analysis.

With regret, I cannot recommend this paper for publication. I do not have complete confidence in the findings.

*Reviewer #3:*

The authors have addressed all the comments to my satisfaction.

---

## [Author Response]

Essential revisions:This is a useful and important analysis of antibiotic usage and resistance patterns across US states. The authors pose a specific question as to whether first use is more closely correlated with resistance than repeat use. A strength is that they are able to explore this for a large number of drug-bug combinations.One obvious difficulty is the definition of first and repeat use. First use per calendar year immediately looks problematic. In the Discussion section, mention is made of first use in a four-year period giving similar results, but there remains the possibility that there was previous usage prior to the observation period. It is likely that some repeat use is misclassified as first use but not vice versa. The scale of the problem could be assessed for, say, years 2, 3 and 4 by checking for usage in the previous calendar year, but this is not done.

We agree that the definitions of “first” and “repeat” present difficulty. Each person has only one truly first use of an antibiotic in their whole lifetime. Any other definition of “first” and “repeat” must be defined with respect to some *ad hoc* time window.

That being said, our goal was to measure the association between the distribution of antibiotic use, at the population level, and antibiotic resistance, at the population level. Our results do not state that individuals who had a recent first use of antibiotics had a higher risk of an antibiotic resistance than individuals with recent non-first use. We only claim that population-level first and repeat use rates have associations with population-level resistance.

Our first-in-that-year and repeat-in-that-year use metrics were intended to characterize the distribution of antibiotic use across a population. We used those metrics because we found them easier to interpret than other measures of the distribution of antibiotic use, say, the Gini coefficient or Shannon entropy.

To clarify this point, we amended the Discussion section to point out that reasoning from the population level to the individual level is difficult because of possible ecological fallacies (subsection “Distribution of antibiotic use and antibiotic resistance”) and because the first-in-a-year and repeat-in-a-year metrics are not obviously sensible at the individual level (subsection “Distribution of antibiotic use and antibiotic resistance”).

Another obvious problem is that the usage data are for outpatients, but the resistance data are (mainly) from hospitals, and that the latter data was only a small sample of the levels of resistance. It's likely that the antibiotics used in hospitals are very different to those used in the outpatient setting and hence the resistances selected may vary. This concern is addressed well in the Discussion section, however, and we don't think there is further work that could be done apart from looking at the subset of community associated infections if those data were available. But this is a major caveat.

We agree that measuring antibiotic resistance at hospitals is a key limitation, and we hope that the caveats in the Results section and Discussion section address this concern. To further emphasize this point, we expanded the note in the Discussion section and included another citation which compares the roles of the hospital and the community in the acquisition of resistant infections.

The key result (subsection “Lack of evidence for more positive association with repeat use”) is that usage-resistance correlations across states tended to be positive for first use and negative for repeat use. The latter result appears to be unexpected and so post hoc explanations are invoked. Care is needed here. It is clearly reasonable to suggest that first use might generate resistance. Presumably subsequent use can then only generate resistance in still susceptible cases (see Introduction). And so on. But it is not clear how that process plays out in terms of generating the correlations explored here. It would have been good to have seen the authors' expectations set out and justified (possibly using a mathematical model). That would help with interpreting the results. Certainly, the Discussion section might benefit from including something about what these conclusions mean for the modelling of resistance. For instance, from a within-host perspective, do the conclusions imply that one dose is enough to generate a resistant dominant population in a host? And that the clearance rate is low?

We heartily agree that these results generate many interesting questions for follow-up. Developing causal explanations that unify the within-host and population-level and explain the observed associations is certainly one of them. However, as we are wary of including more material in a manuscript that introduces a new approach to studying the relationship between antibiotic use and resistance, new metrics, and new datasets, we followed the alternative recommendation and included some points about how these conclusions suggest modeling questions in the Discussion section.

We also do not suspect that repeat use somehow selects for susceptibility. We therefore made revisions that we hope clarify that repeat use tends to have negative regression coefficients only within a multiple regression that also includes first use as a predictor. In other words, repeat use has a negative association with resistance only when first use is controlled for. It is not the case that repeat use has a negative correlation with resistance. This is a subtle but important distinction that we tried to clarify (Results section, Discussion section).

We appreciate the cautionary note about invoking *post hoc* explanations, which, as we understand, refers to the section of the Discussion section that point outs the possibility of “reverse causality” (subsection “Distribution of antibiotic use and antibiotic resistance”). We temper this point by noting that studies with ecological designs are of limited value for distinguishing between different potential causal pathways (subsection “Distribution of antibiotic use and antibiotic resistance”).

More generally, the statistical analyses presented are very simple. This aids transparency, but a lot more could have been looked at with more sophisticated regression modelling (never mind dynamical models). For instance, multivariable, multivariate regression applied to all bug-drug combinations simultaneously could look at drug, extensive/intensive use, bug and geographic (i.e. states) which might be resolvable by combining the data. Such an approach would have also allowed formal model selection/comparison to resolve statistically significant effects. The authors should at least comment on the potential of more advanced analyses and why they chose not to pursue these here.

We agree that this data presents many opportunities for developing and testing more complex methodology to address more sophisticated questions about antibiotic use and resistance. For example, we suspect that use of antibiotic A will select for resistance B in an organism for which resistance to A and B are correlated. Methodology to address these kinds of questions is developing (see citations subsection “Landscape of use-resistance relationships”). Models with more predictors may be more accurate, but we do not have a way to verify them with the available data, and it is not clear they will directly address our central question about the association of the distribution of use with resistance.

That being said, we appreciate this suggestion and amended a section of the Discussion section to propose this more careful approach as a fruitful direction for future work.

In doing more to explain/unravel the trends seen, the route to resistance (e.g. mutation vs. mobile genetic element) might be considered, e.g. exploring if those with a mutational based resistance (e.g. fluoroquinolones) had stronger correlations with use – repeat use in particular.

We agree that this is an interesting idea, but we have two concerns.

First, we expect that the strength of the use-resistance relationship, whether measured by correlation or by regression, will depend on many factors related to the biology and epidemiology of the organism. The resistance mechanism is certainly one of those factors, but it could be that the relationship between quinolone use rates and quinolone resistance among *E. coli* is strong because of factors that have to do with *E. coli* alone or with quinolone use alone, rather than the mechanism of quinolone resistance in *E. coli*. In this sense, we feel the more appropriate hypothesis is that the use-resistance association for a mutational-based resistance to a drug in an organism will be stronger than the association for an HGT-based resistance for the same drug and pathogen. This strikes us as a difficult question that deserves more careful attention and a different data source. In our data sets, resistance is measured phenotypically, not genotypically, so mutational and HGT-based resistance mechanisms cannot be separated for a given drug-organism combination.

Second, we are not entirely sure that our results are consistent with this hypothesis, even when broadly stated. We found macrolides, quinolones, and cephalosporins had systematically more positive use-resistance correlations than nitrofurantoin and TMP/SMX. However, resistance mechanisms do not obviously fall along these lines. Macrolide and nitrofurantoin resistance come from a mix of mutational and HGT-based mechanisms, while cephalosporin resistance is probably plasmid-based. In some cases, quinolone resistance can be HGT-based (Jacoby, 2005). Thus, these distinctions would require a careful review of the dominant resistance mechanisms present in each organism, which is outside the scope of this manuscript.

We would appreciate further suggestions on this point if we have misconstrued the suggestion.

Obviously, this is an analysis of trends in the US. In a few places it would be good to compare the US situation to other countries (e.g. in the percentage of antibiotics used in outpatient vs. healthcare settings, perhaps the distribution of antibiotics used) and to explain more in detail the population (e.g. what percentage of the population is included and what is the Medicare population?).

We thank the reviewers for reminding us to address a global audience. We therefore included:

i) A note with a citation comparing the palette of antibiotics used in the US with those used in other countries (Discussion section), and noting the effect that this might have on the generalizability of our results,

ii) clarified that the statistic about the ratio of outpatient to inpatient antibiotic use is from the UK and Sweden and is merely presumed to also apply to the US (Introduction), and

iii) added an explanation about Medicare when it is first mentioned in the Introduction.

[Editors' note: further revisions were requested prior to acceptance, as described below.]

The manuscript has been improved but we would still like a more effort put into addressing reviewer #2's concerns below regarding hospital vs. outpatient antibiotic use. For instance, presentation of datasets showing some geographic correlation between usage in outpatient and hospital settings.[…]Reviewer #2:I do not think this manuscript has been improved. The authors have responded to the reviewers' comments, but have made only minor changes to the manuscript, even in response to the "essential" revisions. Those changes are largely confined to explanations, caveats and discussion; as far as I can see, they have not revised their analyses in any way.In particular they have not been able to address in any meaningful way the mis-match between scoring antibiotic usage in outpatients but resistance in hospital patients. This seems to me to be a fundamental flaw in the analysis.With regret, I cannot recommend this paper for publication. I do not have complete confidence in the findings.

As we understand, the principal concern is that the antibiotic use data, measured at pharmacies in the community, may not be causally linked to the resistance data, which was collected by hospitals.

To address this concern, in this revision, we also included sensitivity analyses considering changes in the resistance data, as was suggested in the previous round of review: “We don't think there is further work that could be done apart from looking at the subset of community associated infections if those data were available.”

(Sensitivity analyses included in previous revisions only considered the sensitivity of the results to changes in the antibiotic use data.) In a minority of cases, the antibiogram records include annotations indicating whether the isolates whose susceptibility are reported in those records were collected from patients in outpatient facilities, emergency room patients, or inpatients:

67% of records in the ResistanceOpen data are not specific about the origin of the isolates, while the annotations on 20% of records indicate that they refer exclusively to inpatients, and 14% of records indicate that they refer specifically to outpatients and emergency room patients.

In the first new sensitivity analysis, we excluded resistance data from exclusively-inpatient antibiogram records. In the second analyses, we used only data from exclusively-outpatient/emergency room patients. The outpatient-only data is insufficient to achieve statistical significance for the landscape of use-resistance correlations, having at most 21 states per pathogen-drug combination, while our power calculation suggests that at least 33 states are required.

The results of these analyses are shown in

i) Figure 3—figure supplement 1, a new figure showing the correlation landscape for these two sensitivity analyses,

ii) Supplementary file 2 (correlation coefficients for all pathogen-antibiotic-dataset combinations),

iii) Supplementary file 3 (comparison of the sensitivity analyses for the correlation landscape),

iv) Supplementary file 5 (comparison of the sensitivity analyses for the multiple regressions),

v) a new paragraph in subsection “Landscape of correlations between total use and resistance across pathogens and antibiotics”, and

vi) edits to subsection “Lack of evidence for more positive association with repeat use”.

The main conclusions of the paper still hold when using these two subsets of the resistance: use-resistance correlations are more often positive than negative (and not systematically stronger or weaker than the correlations from the complete dataset), and first use regression coefficients are more often positive than repeat use coefficients.

We welcome suggestions for other kinds of analysis that improve the internal validity of these results.

We also wanted to ensure the manuscript is clear about a few points.

First, as mentioned in the previous revision, although most organizations contributing antibiotic susceptibility reports were hospitals, hospital antibiotic susceptibility reports are biased toward community-acquired organisms. The resistance data are “from hospitals”, but this does not mean that the isolates are mostly from hospital-acquired infections (subsection “Landscape of correlations between total use and resistance across pathogens and antibiotics”).

Second, antibiotic resistance in the hospital is not exclusively determined by hospital antibiotic use. The previous revision included a citation for a theoretical study (Knight et al.,2018) about how resistance measured in hospitals, even among inpatients, can be driven by outpatient antibiotic use (subsection “Landscape of correlations between total use and resistance across pathogens and antibiotics”). We supplement that reference with a recently-published one (MacFadden et al., 2018) about how outpatient antibiotic is a driver of ESBL production among *E. coli* in hospitals.

Third, as discussed in subsection “Landscape of use-resistance relationships”, to whatever degree community use is *not* a determinant of antibiotic resistance as measured by hospitals, this mismatch should bias our results toward null conclusions (e.g., correlations not significantly different from zero). In other words, this weakens the power of the study to detect a relationship, rather than increasing the probability that it will discover a false relationship.

We heartily agree with reviewer #2 that there is some mismatch between the use and resistance data, and we do expect this is a limitation to our findings, as laid out in the Introduction, Materials and methods section and Discussion section. As per the above points, we do not consider this mismatch a “fundamental” flaw, but we do invite critiques of the logic laid out above.

References:

Jacoby GA. (2005) Mechanisms of Resistance to Quinolones. Clinical Infectious Diseases, Volume 41, Issue Supplement_2, 15 July 2005, Pages S120–S126, doi: 10.1086/428052